Workshop at the 6th Symposium on Advances in Approximate Bayesian Inference (non-archival), 2024 1–17

# Evaluating approximate Bayesian inference for radio galaxy classification

**Devina Mohan**                    DEVINA.MOHAN@POSTGRAD.MANCHESTER.AC.UK
*Jodrell Bank Centre for Astrophysics*
*Department of Physics & Astronomy*
*University of Manchester, UK*

**Anna M. M. Scaife**[*]                    ANNA.SCAIFE@MANCHESTER.AC.UK
*Jodrell Bank Centre for Astrophysics*
*Department of Physics & Astronomy*
*University of Manchester, UK*

## Abstract

The radio astronomy community is rapidly adopting deep learning techniques to deal with the huge data volumes expected from the next generation of radio observatories. Bayesian neural networks (BNNs) provide a principled way to model uncertainty in the predictions made by such deep learning models and will play an important role in extracting well-calibrated uncertainty estimates on their outputs. In this work, we evaluate the performance of different BNNs against the following criteria: predictive performance, uncertainty calibration and distribution-shift detection for the radio galaxy classification problem.

## 1. Introduction

The radio astronomy community is rapidly adopting deep learning techniques to deal with the huge data volumes expected from the next generation of radio observatories. However, standard neural networks optimise point-wise estimates of the network parameters and provide no uncertainty estimates, which are essential for their scientific applications. On the other hand, Bayesian neural networks (BNNs) provide a principled way to model uncertainty (MacKay, 1992a,b) by specifying priors, $P(\theta)$, over the neural network parameters, $\theta$, and learning the posterior distribution, $P(\theta|D)$, over those parameters, where $D$ is the data. Recovering this posterior distribution directly is intractable for neural networks.

Several approximate inference techniques have been developed to approximate Bayesian inference for neural networks including Hamiltonian Monte Carlo (HMC; Neal and Hinton, 1998; Neal et al., 2011), Variational Inference (VI; Blei et al., 2016; Blundell et al., 2015; Graves, 2011), last-layer Laplace approximation(LLA; Daxberger et al., 2021), MC Dropout (Gal and Ghahramani, 2015) and Deep Ensembles (Lakshminarayanan et al., 2017). Most large scale evaluations of BNNs focus on well-curated terrestrial datasets with lots of labelled examples (Wilson et al., 2022; Vadera et al., 2022). In contrast, in radio astronomy, the largest labelled datasets are of the order $10^3$ (Porter and Scaife, 2023). In this work we present an evaluation of Bayesian deep learning (DL) for radio astronomy, using the morphological classification of radio galaxies as a benchmark.

---

[*] The Alan Turing Institute, London, UK

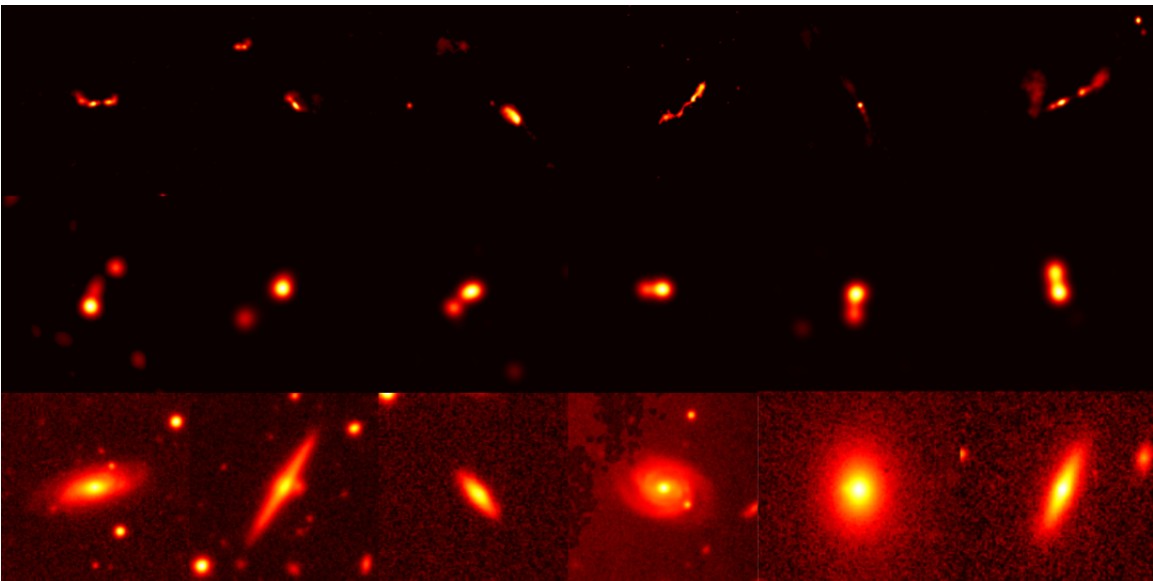

Figure 1: Images from the datasets used in this work. **Top:** FRI/FRII radio galaxies from the MiraBest dataset on which our BNNs are trained on. **Middle:** FRI/FRII galaxies from the MIGHTEE dataset. **Bottom:** Optical galaxies from the GalaxyMNIST dataset. We use the MIGHTEE and GalaxyMNIST datasets to evaluate our models' sensitivity to different types of distribution shifts.

The canonical morphological division of radio galaxies into Fanaroff-Riley Type I (FRI) and Type II (FRII) has persisted as the most common classification scheme for more than 40 years (Fanaroff and Riley, 1974). This scheme has been used to demonstrate improvements in efficiency and accuracy for a variety of deep learning models within both the supervised (Aniyan and Thorat, 2017; Lukic et al., 2019; Becker et al., 1995; Bowles et al., 2021; Scaife and Porter, 2021) and unsupervised (Slijepcevic et al., 2022) learning regimes.

However, recent improvements in the sensitivity and resolution of modern radio astronomy observatories indicate that more complex relationships exist beyond the original FR dichotomy (Mingo et al., 2019). A more nuanced analysis will be enabled by the development of increasingly fine-grained automated classification but the underlying continuum of physical processes that are represented by this diversity of morphology is perhaps better captured by understanding the confidence with which certain galaxies are assigned to different labels by these models. This requires models to focus on uncertainty quantification of model predictions rather than raw predictive performance (Mohan et al., 2022). In this work, we consider the following BNNs for the morphological classification of radio galaxies: HMC, VI, LLA, MC Dropout and Deep Ensembles. The models are evaluated for the following criteria: predictive accuracy, uncertainty calibration and ability to detect different types of distribution shifts.

## 2. Experimental Setup

**Data** We use the MiraBest dataset (Porter and Scaife, 2023) which contains images of FRI/FRII radio galaxies pre-processed to be used for deep learning tasks to train our BNNs. Additionally, to evaluate our models' ability to detect different types of distribution shifts, we consider galaxies from the MIGHTEE (Heywood et al., 2022) and GalaxyMNIST (Walmsley et al., 2022) datasets. MIGHTEE contains images of FRI/FRII radio galaxies from a telescope with different resolution and sensitivity than the MiraBest dataset. GalaxyMNIST contains images of optical galaxies, which exhibit different features and in some sense represent completely out-of-distribution galaxies which well-calibrated models should classify with a very high degree of uncertainty. Details of the datasets used are given in Appendix A. See Figure 1 for examples of galaxies from the three datasets considered in this work.

**Model** We use an expanded LeNet-5 architecture with two additional convolutional layers with 26 and 32 channels, respectively, to be consistent with the literature on using BNNs for classifying the MiraBest dataset (Mohan et al., 2022). The model has $232,444$ parameters in total.

**Inference** To construct the posterior predictive distributions for a single experimental run of VI, LLA and MC Dropout, we obtain $N = 200$ samples from their posterior distributions and calculate $N$ Softmax probabilities for each class, for each galaxy in our test set. For Deep Ensembles we use $N = 10$ samples. For HMC we use the 200 samples obtained after thinning the chains for evaluation. Experimental details for all the inference methods considered in this work are provided in Appendix B. Code for the experiments conducted in this work is available at: https://github.com/devinamhn/RadioGalaxies-BNNs.

**Cold posteriors** Several published works have reported that their BNNs experience a "cold posterior effect (CPE)", according to which the posterior needs to be down-weighted or tempered with a temperature term, $T \leq 1$, in order to get good predictive performance (Wenzel et al., 2020): $P(\theta|D) \propto (P(D|\theta)P(\theta))^{1/T}$. Previous work on using VI for radio galaxy classification has shown that the CPE persists even when the learning strategy is modified to compensate for model misspecification with a second order PAC-Bayes bound to improve the generalisation performance of the network (Mohan et al., 2022; Masegosa, 2019). We do not observe a CPE when we use samples from our HMC inference to construct the posterior predictive distribution for classifying the MiraBest dataset. However, the effect still persists in our VI models. In the general Bayesian DL literature, some authors argue that CPE is mainly an artifact of data augmentation (Izmailov et al., 2021), while others have shown that data augmentation is a sufficient but not necessary condition for CPE to be present (Noci et al., 2021). We find that data augmentation (in the form of random rotations) does not have a significant effect on the cold posterior effect observed in our VI models. However, it does lead to a different degree of trade-off between test error and uncertainty calibration error for our HMC model. The effect of augmentation on performance is further discussed in Section 3.

## 3. Evaluation

### 3.1. Predictive Performance

We use the expected value of the posterior predictive distribution to obtain the classification of each galaxy in the MiraBest test set and calculate the test error for a single experimental run by taking an average of the classification error over the entire test set. We report the mean and standard deviation of the test error for 10 experimental runs, see Table 1.

VI has the best predictive performance, irrespective of whether data augmentation is used or not. The low standard deviation values for VI indicate that the mean of the posterior predictive distribution found by VI optimisation is robust to random seeds and shuffling. The same does not hold true for LLA and Dropout, which are the two worst performing models. Deep ensembles lie somewhere in between. The MAP value reported in Table 1 is chosen on the basis of the lowest validation loss from the ensemble of CNNs that we trained.

### 3.2. Uncertainty Calibration

We report the expected uncertainty calibration error (UCE; Gal and Ghahramani, 2015; Laves et al., 2019; Mohan et al., 2022) of the predictive entropy of our posterior predictive distributions in Table 1. The uncertainty quantification metrics are recalled in Appendix C. For HMC, VI, LLA and MC Dropout, we use the 64% credible intervals of the posterior predictive distributions to calculate UCE. For Deep Ensembles, we use the entire posterior predictive distribution constructed using the 10 ensemble members.

We find that HMC without data augmentation is the most well-calibrated BNN for the radio galaxy classification problem. HMC with data augmentation has a higher UCE. VI models with and without data augmentation are similarly calibrated. The high standard deviation values show how sensitive VI is to initialisation, and this is a well documented issue in the literature (Altosaar et al., 2018; Rossi et al., 2019). LLA, MC Dropout and Deep Ensembles are very poorly calibrated compared to HMC and VI.

We refrain from reporting the mutual information and conditional entropy as measures of epistemic and aleatoric uncertainty since they are known to be dependent on model specification and class separability (Hüllermeier and Waegeman, 2021), making them difficult to interpret given our small statistical sample of radio galaxies. More recently, Wimmer et al. (2023) have also shown that the additive decomposition of total predictive uncertainty into mutual information and conditional entropy breaks down in machine learning settings where we have access to a limited number of data samples.

### 3.3. Detecting Distribution Shift

The independent and identically distributed (i.i.d.) assumption breaks down when neural networks are deployed on real-world datasets. This can be due to a change in the input data distribution, for instance when the model is faced with galaxies from a new telescope facility, leading to covariate shift. Some degree of semantic shift is also expected when the distribution of labels changes at test time due to the presence of new classes when telescopes with improved resolution and sensitivity reveal new morphologies of galaxies. For example, the MIGHTEE survey revealed previously unresolved diffuse emission which led to the discovery of new giant radio galaxies (Delhaize et al., 2020). Our application also presents

Table 1: Test error and uncertainty calibration error (UCE) of the predictive entropy for all the Bayesian neural networks considered in this work. We also provide a baseline MAP error percentage. Inference methods with a (*) indicate that no data augmentation was used during inference for those experiments.

| Inference | Error (%) ↓ | UCE ↓ |
|---|---|---|
| HMC | $4.16 \pm 0.45$ | $14.76 \pm 0.95$ |
| HMC* | $6.24 \pm 0.45$ | $12.65 \pm 0.01$ |
| VI | $3.94 \pm 0.01$ | $12.77 \pm 6.11$ |
| VI* | $3.84 \pm 0.01$ | $12.32 \pm 6.36$ |
| LLA | $8.85 \pm 2.09$ | $23.84 \pm 3.54$ |
| Dropout | $7.88 \pm 2.81$ | $25.75 \pm 4.44$ |
| Ensembles | $7.69 \pm 0.27$ | $24.41$ |
| MAP | $5.76$ | |

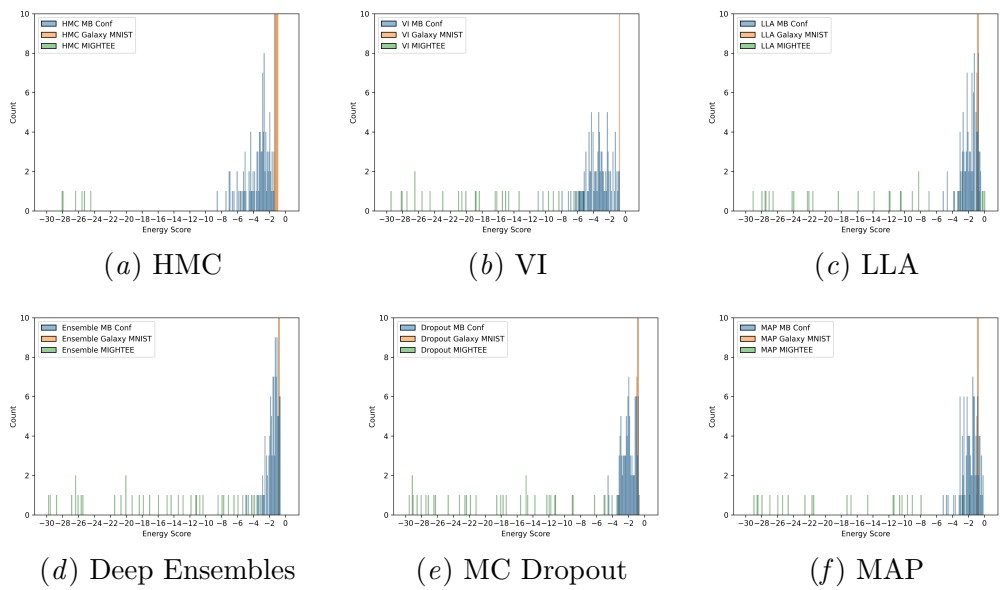

Figure 2: Histograms of energy scores calculated for the MiraBest (MBConf; blue), GalaxyMNIST (orange) and MIGHTEE (green) test datasets for the different models considered in this work, see Section 3.3 for details.

a unique conundrum where defining what out-of-distribution is difficult because the galaxies exist on a spectrum of physical processes that give rise to different astrophysical phenomena and different morphologies. Thus we expect there to be a combination of different types of distribution shifts when BNNs are deployed in radio astronomy.

Liu et al. (2020) propose a post-hoc scoring function for discriminative classification models which can be used to distinguish between in-distribution (iD) and distribution-shifted data examples. We calculate average scalar energy scores for different test samples, $x$, for all the datasets considered in this work using the logit values, $f_i(x)$, for each class, $i$, using $N$ posterior samples:

$$\tilde{\mathrm{E}}(x; f) = \frac{1}{N} \sum_j^N -T.\log \sum_i^K e^{f_i(x)/T}, \tag{1}$$

where the temperature term, $T$, is set to 1. Out-of-distribution (OoD) samples are expected to have higher energy in this framework.

Histograms of energy values for the different inference methods considered in this work are shown in Figure 2. We use the models with the lowest validation error from the experiments to calculate the energy scores. The iD MiraBest samples get mapped to a larger interval of energy values by our HMC and VI models. In comparison, the energy scores for iD samples lie in a very narrow interval for Deep Ensembles, MC Dropout and LLA, which suggests that fewer iD samples have been pushed to lower energy values.

The FRI/FRII galaxies from the MIGHTEE dataset present a significant dataset shift due to differences in observational properties. MIGHTEE galaxies get mapped to a large interval of energy values, in some cases extending upto $E = -90$. However, HMC is the only model for which there exists a clear distinction between iD FRI/FRII galaxies from MiraBest and distribution-shifted FRI/FRII galaxies from MIGHTEE.

We find that HMC and VI models are good at separating the OoD optical galaxies from the GalaxyMNIST dataset, see Figure 2(a) and Figure 2(b). For all other models, there is a significant degree of overlap between the iD and OoD samples, see Figure 2. We also note that LLA maps some of the MIGHTEE galaxies to energies higher than OoD GalaxyMNIST data, see Figure 2(c).

## 4. Discussion

A certain degree of trade-off exists between a model's predictive performance and calibration. While VI has the best predictive performance, HMC without data augmentation is the most well-calibrated model and only 2.5% less accurate. HMC with data augmentation has a better predictive performance, but is less calibrated than HMC without data augmentation. A similar trade-off has also been reported by Krishnan and Tickoo (2020), who propose a loss function which optimises for both accuracy and calibration.

The differences in dataset separability via energy scores for different BNNs can be better understood if we examine the way in which each of these models is being optimised. LeCun et al. (2006) show that many modern learning algorithms can be interpreted as energy-based models. In the energy-based framework, different loss objectives cause certain inputs' energies to be pulled up/down. LLA, Deep ensembles and MC Dropout are all trained by minimising the negative log likelihood (NLL) loss plus some regularisation term due to weight decay. Our evaluation suggests that NLL training is not be able to shape the energy functional well enough to distinguish between the datasets we have considered. While HMC is directly sampling from an energy surface that is proportional to the log of the posterior

distribution, is case of VI the Evidence Lower Bound (ELBO) provides a well optimised surrogate energy function. Our HMC and VI models seem to have learned good energy surfaces. LeCun et al. (2006) also note that softmax probabilities can be considered good if the energy function is estimated well enough from the data. Perhaps this is also why HMC and VI are the better calibrated models among all those we have considered in this work.

Our observations on the cold posterior effect (CPE) contradict the results presented in Izmailov et al. (2021). They suggest that the CPE is largely due to data augmentation. While our HMC model does not require any tempering, the VI models require temperatures below $T = 0.01$ to produce good predictive performance. We also found that data augmentation does not have a significant impact on the CPE observed in our models. Finding the cause of the cold posterior effect observed in VI for radio galaxy classification is still an open research question.

While Deep Ensembles are generally considered a good approximation to the Bayesian posterior, Seligmann et al. (2023) recently showed that single-mode BDL algorithms approximate the posterior better than Deep Ensembles. We also find that Deep Ensembles do not work as well as VI and HMC for our application. Thus we find that results from the CS literature where models are trained on terrestrial datasets often do not translate to domain-specific applications.

## 5. Conclusions

In this work we have evaluated different Bayesian neural networks for the classification of radio galaxies. We found that Hamiltonian Monte Carlo and variational inference perform well at our model and dataset scales for the three criteria we considered: predictive performance, uncertainty calibration and ability to detect distribution shift. Commonly used Bayesian NNs such as MC Dropout and Deep Ensembles are poorly calibrated for our application. Since HMC is very computationally heavy, optimising VI for future radio surveys might be the way forward.

## Acknowledgments

AMS gratefully acknowledges support from an Alan Turing Institute AI Fellowship EP/V030302/1.

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

## Appendix A. Data

Radio galaxies are characterised by large scale jets and lobes which can extend up to mega-parsec distances from the central black hole and are observed in the radio spectrum. The original binary classification scheme proposed to classify such extended radio sources was based on the ratio of the extent of the highest surface brightness regions to the total extent of the galaxy (Fanaroff and Riley, 1974). FRI galaxies are edge-darkened whereas FRII galaxies are edge-brightened. Over the years, several other morphologies such as bent-tail (Rudnick and Owen, 1976; O'Dea and Owen, 1985), hybrid (Gopal-Krishna and Wiita, 2000), and double-double (Schoenmakers et al., 2000) sources have also been observed and there is still a continuing debate about the exact interplay between extrinsic effects, such as the interaction between the jet and the environment, and intrinsic effects, such as differences in central engines and accretion modes, that give rise to the different morphologies.

We train our BNNs on the MiraBest Confident dataset [Section A.1] and use the MIGHTEE [Section A.2] and GalaxyMNIST [Section A.3] datasets to test the ability of our BNNs to detect different types of distribution shifts.

### A.1. MiraBest

The MiraBest dataset used in this work consists of 1256 images of radio galaxies of $150 \times 150$ pixels pre-processed to be used specifically for deep learning tasks (Porter and Scaife, 2023). The galaxies are labelled using the FRI and FRII morphological types based on the definition of Fanaroff and Riley (1974) and further divided into their subtypes. In addition to labelling the sources as FRI, FRII and their subtypes, each source is also flagged as 'Confident' or 'Uncertain' to indicate the human classifiers' confidence while labelling the dataset. In this work we use the MiraBest Confident subset and consider only the binary FRI/FRII classification during training, see Figure 1 (top row) for some examples. The training and validation sets are created by splitting the predefined training data into a ratio of 80:20. The final split consists of 584 training samples, 145 validation samples, and 104 withheld test samples.

The MiraBest dataset was constructed using the sample selection and classification described in Miraghaei and Best (2017), who made use of the parent galaxy sample from Best and Heckman (2012). Optical data from data release 7 of Sloan Digital Sky Survey (SDSS DR7; Abazajian et al., 2009) was cross-matched with NRAO VLA Sky Survey (NVSS; Condon et al., 1998) and Faint Images of the Radio Sky at Twenty-Centimeters (FIRST; Becker et al., 1995) radio surveys.

### A.2. MIGHTEE

The MIGHTEE dataset is constructed using the Early Science data products from the MeerKAT International GHz Tiered Extragalactic Exploration survey (MIGHTEE; Heywood et al., 2022). MIGHTEE is an ongoing radio continuum survey being conducted using the MeerKAT telescope, which is one of the precursors to the Square Kilometer Array (SKA). The survey provides radio continuum, spectral line and polarisation data, of which we use the radio continuum data and extract images for the COSMOS and XMMLSS fields. While there are thousands of objects in these fields, expert labels are only available

for 117 objects. We use the data pre-processing and expert labels made available by Slijepcevic et al. (2024). The dataset contains classifications based on the consensus of five expert radio astronomers. The final sample contains 45 FRI and 72 FRII galaxies, see Figure 1 (second row). We note that the MIGHTEE dataset contains significant observational differences from the MiraBest dataset.

### A.3. Galaxy MNIST

In addition to considering different datasets of radio galaxies which have been curated using using data from radio telescopes, we also evaluate our models on data collected from optical telescopes. Optical images of galaxies contain different features and in a sense represent completely out-of-distribution galaxies which well-calibrated models should classify with a very high degree of uncertainty so that they can be flagged for inspection by an expert.

We use the GalaxyMNIST[1] dataset which contain images of $10,000$ optical galaxies classified into four morphological types using labels collected by the Galaxy Zoo citizen science project, see Figure 1 (last row) for examples. The galaxies are drawn from the Galaxy Zoo Decals catalogue (Walmsley et al., 2022). We resize the high resolution images from 224x224 to 150x150 to match the input dimensions of our model. We construct a small test set of 104 galaxies from the dataset to evaluate the out-of-distribution detection ability of our BNNs.

## Appendix B. Experimental Details

The experimental details for all the Bayesian neural networks considered in this work are presented in this section. The networks are trained on the MiraBest dataset.

### B.1. HMC Inference

The first application of MCMC to neural networks was proposed by Neal and Hinton (1998), who introduced Hamiltonian Monte Carlo (HMC) from quantum chromodynamics to the general statistics literature. However, it wasn't until Welling and Teh (2011) introduced Stochastic Gradient Langevin Dynamics (SGLD), that MCMC for neural networks became feasible for large datasets. More recently, Cobb and Jalaian (2021) have revisited HMC and proposed novel data splitting techniques to make it work with large datasets. We use the HMC algorithm in our work.

We use the HAMILTORCH package[2] developed by Cobb and Jalaian (2021) for scaling HMC to large datasets. Using their HMC sampler, we set up two HMC chains of $200,000$ steps using different random seeds and run it on the MiraBest Confident dataset. We use a step size of $\epsilon = 10^{-4}$ and set the number of leapfrog steps to $L = 50$. We specify a Gaussian prior over the network parameters and evaluate different prior widths, $\sigma = \{1, 10^{-1}, 10^{-2}, 10^{-3}\}$, using the validation data set. We find that $\sigma = 10^{-1}$ results in the best predictive performance and consequently use it to define the prior width for all weights and biases of the neural network in our experiments. To compute the final posteriors we thin the chains by a factor of 1000 to reduce the autocorrelation in the samples and obtain

---

1. https://github.com/mwalmsley/galaxy_mnist
2. https://github.com/AdamCobb/hamiltorch

200 samples. A compute time of 170 hrs is required to run the inference on two Nvidia A100 GPUs. The acceptance rate of the proposed samples is 97.62%. We repeat the inference with data augmentation (random rotations).

**Assessing Convergence**: The Gelman-Rubin diagnostic, $\hat{R}$, is used to assess the convergence of our HMC chains (Gelman and Rubin, 1992). If $\hat{R} \approx 1$ we consider the HMC chains for that particular parameter to have converged. We examine the convergence of the last layer weights and find that using data augmentation leads to a higher proportion of weights with $\hat{R} \geq 1$. We also monitor the negative log-likelihood and accuracy, which converge by the $100,000^{\text{th}}$ inference step.

## B.2. Other inference methods

We conduct 10 experimental runs for each inference method presented in this section using different random seeds and random shuffling of data points between the training and validation datasets.

### B.2.1. Variational Inference

Variational inference (VI) assumes an approximate posterior from a family of tractable distributions, and converts the inference problem into an optimisation problem. The model learns the parameters of the distributions by minimising an Evidence Lower Bound Objective (ELBO) function, which is composed of a data likelihood cost and a complexity cost which quantifies the difference between the prior and the variational approximation using KL divergence. We use the Bayes by Backprop (BBB) algorithm (Blundell et al., 2015) in this work.

We make a Gaussian variational approximation to the posterior and find that our model is optimised with a Gaussian prior width $\sigma = 0.01$. We also test a Laplace prior following (Mohan et al., 2022), but find that it does not lead to a significant performance improvement. Results are reported for a tempered VI posterior, with $T = 0.01$. The network is trained for 1500 epochs using the Adam optimser with a learning rate of $5.10^{-5}$. A compute time of 40 mins is required to train the VI model on a single Nvidia A100 GPU.

### B.2.2. Last Layer Laplace Approximation

Last-layer Laplace approximation (LLA) constructs Gaussian approximations around the MAP values learned by standard NN training using the second order partial derivatives of the loss function, $\mathcal{L}$ Daxberger et al. (2021). This method allows one to learn posteriors for the last layer weights of the network, $\theta^{(L)}$, while keeping the rest of the values fixed at their MAP estimates. The covariance matrix for the last layer is calculated using the empirical Fisher approximation to the Hessian, which contains information about the local curvature of the loss function for each parameter. The method assumes a zero mean Gaussian prior $p(\theta) = \mathcal{N}(\theta; 0, \gamma^2 I)$. The prior variance, $\gamma^2$, is estimated using marginal likelihood maximisation (Immer et al., 2021; Daxberger et al., 2021).

We use the MAP values learned by our non-Bayesian CNNs to construct our last-layer Laplace approximation using the Laplace package[3] developed by Daxberger et al. (2021).

---

3. https://github.com/AlexImmer/Laplace

We use a diagonal factorisation of the Hessian. The optimised prior standard deviation found using marginal likelihood maximisation for 10 experimental runs lies between $\sigma \in [0.03, 0.04]$.

### B.2.3. Deep Ensembles

One can also use the output of multiple randomly initialised models to form a uniformly-weighted mixture model whose predictions can be combined to form an ensemble (Lakshminarayanan et al., 2017). We train 10 non-Bayesian CNN models with different random seeds and randomly shuffled training:validation splits to construct the posterior predictive distribution by combining the softmax values obtained for each galaxy in our test set. The models are trained for 600 epochs using the Adam optimiser with a learning rate of $10^{-4}$ and weight decay $10^{-6}$. We use a learning rate scheduler which reduces the learning rate by 10% if the validation loss does not improve for two consecutive epochs and use an early stopping criterion based on the validation loss.

### B.2.4. MC Dropout

Another easily implemented Bayesian approximation is MC Dropout, which learns a distribution over the network outputs by setting randomly selected weights of the network to zero with probability, $p$ (Gal and Ghahramani, 2015). MC dropout can be considered an approximation to VI, where the variational approximation is a Bernoulli distribution.

A dropout rate of 50% is implemented before the last two fully-connected layers of our neural network. This dropout configuration performed better compared to implementing dropout only before the last layer of the network. The network is trained for 600 epochs using the Adam optimser with a learning rate of $10^{-3}$ and a weight decay of $10^{-4}$. We use a learning rate scheduler which reduces the learning rate by 10% if the validation loss does not improve for two consecutive epochs and use an early stopping criterion based on the validation loss.

## Appendix C. Evaluation metrics

### C.1. Predictive entropy

Using Monte Carlo (MC) samples obtained from the posterior predictive distributions of different BNNs, one can obtain $N$ Softmax probabilities for each class, $c$, in the dataset. We can recover $N$ class-wise Softmax probabilities as follows:

$$P(y|x, D) = \frac{1}{N} \sum_{i=1}^{N} P(y = c|x, w^{(i)}), \tag{2}$$

where $w^{(i)}$ is the $i^{\text{th}}$ weight sample from the posterior distribution conditioned on the training data, $D$, and $(x, y)$ are data from the test set. Using these samples, one can quantify the uncertainty in the predictions using different metrics. In this work we look at the predictive entropy of the softmax distribution which measures the average amount of information inherent in the distribution and is defined as:

$$\mathbb{H}(y|x, D) = -\sum_{c} P(y = c|x, w) \log P(y = c|x, w), \tag{3}$$

which can be approximated using MC samples as follows (Gal, 2016):

$$\mathbb{H}(y|x, D) = -\sum_c \left( \frac{1}{N} \sum_{i=1}^N P(y = c|x, w^{(i)}) \right) \log \left( \frac{1}{N} \sum_{i=1}^N P(y = c|x, w^{(i)}) \right). \tag{4}$$

### C.2. Uncertainty Calibration Error (UCE)

To examine how well calibrated the preditive entopy is, we calculate the Uncertainty Calibration Error (UCE), which is a more general form of the Expected Calibration Error (ECE). UCE is a weighted average of the difference between fractional error and uncertainty calculated for the output of the model when binned into $M$ bins of equal width for a particular uncertainty metric:

$$\text{UCE} = \sum_{m=1}^M \frac{|B(m)|}{n} |\text{err}(B_m) - \text{uncert}(B_m)|. \tag{5}$$

Here $B_m$ is the set of data in a particular bin, $n$ is the total number of data points and uncert$(B_m)$ is the average value of a given uncertainty metric for those data points:

$$\text{uncert}(B_m) = \frac{1}{|B_m|} \sum_{i \in B_m} \text{uncert}_i , \tag{6}$$

where uncert$_i$ can be calculated using Equation 4 followed by minmax-normalisation to bring values into the range 0 to 1.

Equation 16 of Laves et al. (2019) defines the average fractional error in bin $B_m$ to be:

$$\text{err}(B_m) = \frac{1}{|B_m|} \sum_{i \in B_m} \text{err}_i , \tag{7}$$

where err$_i$ is the contribution to this error from an individual data point, defined as:

$$\text{err}_i = \mathbf{1}(\hat{y}_i \neq y) \quad \forall \ i \in B_m . \tag{8}$$

Here we redefine err$_i$ to be the average error obtained for an individual data sample, such that

$$\text{err}_i = \frac{1}{N} \sum_{j=1}^N \mathbf{1}(\hat{y}_{ij} \neq y) \quad \forall \ i \in B_m , \tag{9}$$

where $N$ is the number of samples drawn from the posterior predictive distribution.

