# OpenReview forum: "Evaluating approximate Bayesian inference for radio galaxy classification"
_approximateinference.org/AABI/2024/Symposium — AABI 2024_

### Official Review · Reviewer_ir7r · 2024-04-08
**The review outlines datasets like MiraBest, MIGHTEE, and Galaxy MNIST, and evaluates Bayesian neural network inference methods including HMC, VI, LLA, Deep Ensembles, and MC Dropout, emphasizing robust evaluation methodologies for radio galaxy classification tasks.**

**Rating:** 6
**Confidence:** 4

**Review:**

Areas of improvements:
1. Further exploration of the interpretability of uncertainty estimates generated by different inference methods would enhance the paper's practical utility. Understanding the sources and implications of uncertainty in classification results is crucial for informed decision-making in astrophysical analyses.
2. The study could expand its discussion to explore broader implications for astrophysical research and the broader scientific community. This broader context would enhance the relevance and impact of the work.
3. More detailed discussion on potential limitations encountered during the experiments, such as computational constraints, model complexity, or dataset biases would be encouraged. Addressing these limitations would provide a more nuanced understanding of the proposed methods' practical application.

Pros:
1. The study provides a thorough assessment of various Bayesian inference methods, including Hamiltonian Monte Carlo and variational inference, for the classification of radio galaxies.
2. By considering multiple datasets such as MiraBest, MIGHTEE, and GalaxyMNIST, the research offers a comprehensive comparison of Bayesian neural networks' performance under different data distributions. This ensures the robustness and generalizability.
3. The application of Bayesian neural networks to radio galaxy classification represents a novel and innovative approach in the field of astrophysics. By leveraging advanced ML techniques, the study opens up new avenues for automated classification and analysis of complex astronomical data.

---

### Official Review · Reviewer_94uz · 2024-04-13
**Review of "Evaluating approximate Bayesian inference for radio galaxy classification"**

**Rating:** 7
**Confidence:** 4

**Review:**

This paper evaluates the performance of Bayesian inference methods for radio galaxy classification, focusing on the challenges and benefits of incorporating uncertainty estimates in deep learning models for radio astronomy. Various inference techniques, including Variational Inference and Hamiltonian Monte Carlo, are compared and analyzed in the context of radio galaxy classification tasks.
I like that this paper presents a practical application of Bayesian deep neural networks for radio astronomy image classification. The authors do a good job comparing several Bayesian methods and explaining their strengths and weaknesses.
The neural networks in this study are quite standard and basic but probably adequate for such a task.
However, they ONLY evaluate Bayesian methods. The absence of a comparison with traditional (frequentist?!) neural networks in the evaluation of Bayesian deep neural networks could be considered a weakness in the study. A comparison with traditional neural networks would provide a benchmark for assessing the effectiveness of Bayesian approaches in terms of predictive performance, uncertainty calibration, and other criteria. Such a comparison could offer insights into the added value of Bayesian methods in the specific context of radio galaxy classification and help validate the benefits of incorporating uncertainty estimates in deep learning models for radio astronomy.
Still, the paper is well-written with no apparent typos or grammatical errors. The language used is technical and precise and the authors communicated their research findings and methodology in a clear and concise manner.
I like that the paper is very accessible and provides a real-work use case.

---

### Official Review · Reviewer_rSvY · 2024-04-22

**Rating:** 8
**Confidence:** 4

**Review:**

This paper evaluates the performance of various BNN methods on radio galaxy data. Unlike the observation on CIFAR-10, this paper observes that the cold posterior effect is much less pronounced. In addition, VI shows great calibration and predictive accuracy in this setting, whereas deep ensembles and LLA, which shows great performance in CIFAR-10 and other standard academic datasets, shows significant worse test error and calibration in galaxy data. This is a great example illustrating the importance of entropy regularization introduced by HMC (implicitly) and VI (explicitly).

Overall, I find this paper of great value for the Bayesian deep learning community, in particular, it shows that, although BNN on CIFAR-10 already seems to be a dead and meaningless task (deep ensemble always wins), on many real world noisy datasets, with limited amount of annotations, there are still many advantages and unsolved problems for BNN.

---

### Official Review · Reviewer_LWQB · 2024-04-24

**Rating:** 8
**Confidence:** 3

**Review:**

I think this is a great submission! A very interesting read and super well written.

I really like careful and thorough comparisons of methods on domain-specific applications. It would be great to see more paper like this.

I did not find any major shortcomings. I just got one question and one comment:

* Why do you choose the 64% credible intervals for evaluation? I can imagine that the results become more drastic for e.g. the 99% interval, where all methods but HMC will likely fall over. It would be interesting to show the UCE for a number of credible intervals: 65%, 90%, 95%, 99%.

* Accessibility of the exposition would be improved by explaining by you expect lower energy for OOD samples. One simple explanation could be something along these lines: the energy is the average negative log-normalisation constant of the predictive distribution, which tends to be low at points where the density of the distribution is low.

---

### Official Review · Reviewer_WDkf · 2024-04-24
**A well written article and interesting benchmark**

**Rating:** 7
**Confidence:** 2

**Review:**

This is an interesting comparison of Bayesian Deep Learning (BDL) approaches which appears very well executed. I appreciated the variety of the metrics and their relevance, in particular the evaluation of uncertainty calibration. I think it is valuable to show that BDL approximate methods can be implemented and compared on relevant tasks. In this regard, it would be precious to have access to the replication material to see and possibly re-use the authors' implementation.

For readability and self-containment, the metrics used in the paper could be recalled, possibly in the supplementary material.

The data augmentation scheme is not described.

It would also be interesting to discuss a little more potential differences between the existing benchmarks and this one, to shed more light on why conclusions may differ.

Side note: At the end of the first paragraph, it is written that recovering the posterior distribution directly is intractable. Then, approximate methods are detailed, but Hamiltonian Monte Carlo comes much later and is not presented as an "exact up to MC error" method, although it seems to be considered quite like the gold standard in the rest of the paper. It would probably be more instructive for the reader to present HMC along with the approximate methods, mentioning that it is not approximate in the same sense as the other since the approximation errors could decrease to 0 with increasing computation time.

---

### Official Review · Reviewer_DmHQ · 2024-04-25
**Nice experimental results but insufficient new contributions**

**Rating:** 4
**Confidence:** 3

**Review:**

The manuscript presents evaluation of different Bayesian neural networks for the classification of radio galaxies. Specifically, three evaluation criteria: predictive performance, uncertainty calibration and ability to detect distribution shift were quantified. The detailed comments on this work are as follows:

Pros :

1) The manuscript is nicely written and is easy to follow. The problem statement, methodology, experiments (datasets and implementation) have been presented clearly.

Cons :

1) The manuscript presents an experimental analysis of existing BNNs on well known evaluation metrics. Experimental analysis cannot be considered as a 'Significant Contribution'.

2) Performance analysis was described in the text. Mathematical reasoning inline to the specific task (radio galaxy classification) on the performance of BNNs would have been interesting.

---

### Meta-Review · Area_Chair_zWb5 · 2024-05-12

**Recommendation:** Accept (Poster)
**Confidence:** 4

**Metareview:**

The paper provides a benchmark of different BNN methods in the domain of astronomy, specifically for radio galaxy classification. This is an important topic to drive the Bayesian deep learning field forward in terms of its applications in broader science. The paper shows that HMC and variational inference might be better compared to other popular methods such as deep ensemble.

I recommend acceptance.

However, I strongly encourage the authors to address all reviewers' comments. Moreover, in the submission version of the paper, the abstract is missing---it should be added in the final version.

---

### Decision · Program_Chairs · 2024-05-27

Accept